# STEELNET: MULTIMODAL REPRESENTATION LEARNING FOR INDUSTRIAL PROCESS OPTIMIZATION

## ABSTRACT

Steel rolling mills must continuously monitor various sensors like vibration probes, thermocouples, and flow meters to ensure safe operations and maintain product quality. However, harsh industrial conditions sometimes lead to sensor failures, unclear signals, and incomplete data. Traditional monitoring systems often struggle in these environments, making it challenging to detect early signs of problems and predict failures. To this end, we propose SteelNet: a multimodal representation learning framework designed for robust learning from various industrial sensor data. SteelNet incorporates cross-modal alignment and modality dropout strategies that enable consistent representation learning even when modalities are partially missing. The core problem being solved is improving equipment availability and optimizing process parameters. This framework allows for early detection of critical events by combining information from multiple sensors and effectively handling missing data, which are common in industrial environments. By improving the reliability of anomaly detection and predictive insights, SteelNet not only strengthens fault tolerance but also supports better decision-making in process optimization. Although developed for steel rolling mills, its applicability extends to real-world scenarios and other industrial setups.

## 1 INTRODUCTION

Steel rolling mills are vital to global manufacturing, providing materials for construction, transport, and energy systems. As complex cyber-physical plants, they require continuous monitoring of diverse sensors such as vibration probes, thermocouples, flow meters, acoustic devices, and cameras to ensure safe operations, equipment availability, and product quality. Yet harsh environments often cause noisy signals, malfunctions, or missing data, complicating predictive maintenance and anomaly detection, and increasing unplanned downtime risks (Rao, 2024; Rahman et al., 2025). With Industry 4.0 and industrial IoT, the surge in multimodal sensor data needs more robust methods to integrate non-similar inputs for fault diagnosis and process optimization (Chen, 2022; Wang et al., 2018).

Extensive research has applied multimodal deep learning to predictive maintenance. Rao (2024) combined CNNs and LSTMs for imagery and time-series data, while Rahman et al. (2025) proposed MultiSenseNet using CNNs, transformers, and GNNs for failure prediction. Other studies fused categorical and text inputs (Lichtenwalter et al., 2021), applied reinforcement learning for adaptive control (Liu et al., 2024), and showed that multimodal knowledge integration improves maintenance and anomaly detection (Shen et al., 2024; McKinney et al., 2025).

Surveys have synthesized advances in multimodal intelligence and representation learning. Zhang et al. (2020) reviewed fusion techniques for image-text tasks, while Guo et al. (2019) examined joint, coordinated, and encoder-decoder approaches. Wang & Yue (2012) focused on trends in manufacturing, highlighting inter-modal, intra-modal, and domain knowledge fusion. At the process level, Wu & Liang (2024) discussed knowledge acquisition and representation for intelligent manufacturing. Benchmarks like MultiBench (Liang et al., 2021) stress evaluating generalization and robustness to missing modalities. Collectively, these works provide the foundation for multimodal industrial applications.

Recent advances highlight cross-modal architectures for anomaly detection. Wu et al. (2024) introduced FmFormer, a transformer for video and current signals in smelting, while Kong et al. (2025)

proposed multi-modal, multi-level features for flow pattern identification in oil-water pipelines. In manufacturing forecasting, Zurita et al. (2017) used adaptive-neurofuzzy models, and Liu et al. (2021) developed a stacked multimanifold autoencoder for industrial data. Multimodal sensing has also improved machining quality prediction (Sheng et al., 2024) and flaw detection in additive manufacturing (Petrich et al., 2021), collectively underscoring the value of multimodal feature extraction for industrial reliability.

In steelmaking, multimodal learning is widely applied. Song et al. (2019) used CNN+DNN for temperature control in continuous casting, and Lee et al. (2020) employed CNN+RNN for transient temperature prediction. Zhang et al. (2023a) studied slag inclusion prediction, while Peng et al. (2022) proposed big data-driven methods for hot rolling performance. Reinforcement learning has been used for process optimization (Liu et al., 2023) and surface defect detection (Zhang et al., 2023b). Knowledge graphs aid product development (Peng et al., 2024) and multimodal fault diagnosis (Wu et al., 2023), with Industry 4.0 data mining frameworks integrating heterogeneous sources for fault detection (Chen, 2022).

Explainable and knowledge-driven approaches enhance multimodal learning. Calaon et al. (2024) combined multimodal predictive analysis with explainable AI for root cause analysis, while Wang et al. (2025) developed an LLM-based framework for intelligent perception and decision-making. Liang et al. (2025) proposed fine-grained multimodal reasoning for product design. Knowledge graphs support defect reasoning (Zhang et al., 2023b) and metallurgical knowledge reuse (Peng et al., 2024), highlighting the importance of robustness, interpretability, and decision support in industrial multimodal systems.

Despite advances, two challenges persist: robustness to missing or corrupted modalities, and aligning predictions with actionable metrics like equipment availability. While frameworks such as MultiSenseNet (Rahman et al., 2025), FmFormer (Wu et al., 2024), and unsupervised fusion methods (McKinney et al., 2025) perform well under ideal conditions, real-world plants face modality dropout, occlusion, and noise. Few methods explicitly optimize equipment uptime, a key industrial metric.

### CONTRIBUTION OF THIS WORK

This paper introduces **SteelNet**, a multimodal representation learning framework tailored for steel rolling mills. SteelNet employs cross-modal alignment to integrate heterogeneous sensor modalities into a shared space with close mapping together and modality-dropout training to sustain performance under missing data. By using anomaly detection with predictive insights together, SteelNet directly targets equipment availability, thereby improving both fault tolerance and operational decision-making. While motivated by steel rolling mills, the framework generalizes to other industrial domains where multimodal, imperfect, and heterogeneous data are the norm.

## 2 METHOD

### 2.1 DATASET

Our experimental evaluation is conducted on a novel multimodal dataset constructed specifically to address causal inference and process optimization challenges in steel rolling mills. This dataset extends the Severstal Steel Defect Detection competition data (Grishin et al., 2019) by incorporating synthetically generated process parameters that simulate realistic industrial monitoring scenarios.

#### 2.1.1 BASE DATASET AND DEFECT INTENSITY SCORING

The foundation of our dataset originates from the Severstal Steel Defect Detection competition (Grishin et al., 2019), which contains 12,568 high-resolution steel surface images with pixel-level annotations for four defect classes:

- Class 1: Crazing (thermal stress patterns) - 897 instances
- Class 2: Inclusion (contamination particles) - 247 instances
- Class 3: Patches (coating irregularities) - 5,150 instances

- Class 4: Pitted Surface (corrosion damage) - 801 instances

To enable multimodal learning that incorporates defect severity information, we developed a deep learning-based defect intensity scoring system. Using a ResNet-34 backbone with a regression head, we trained a model to predict per-class defect fractions by regressing pixel-level defect coverage from the original RLE-encoded segmentation masks. The model was trained for 8 epochs using MSE loss, achieving a validation loss of 0.003261. For each image, the defect intensity score is computed as a weighted combination of the predicted per-class fractions:

$$\text{intensity\_score} = \sum_{c=1}^{4} w_c \cdot f_c \tag{1}$$

where $f_c$ represents the predicted fraction of pixels belonging to defect class $c$, and $w_c = [1.0, 1.5, 2.0, 1.2]$ are empirically determined weights that reflect the relative severity of each defect type in industrial contexts.

### 2.1.2 PROCESS PARAMETER AUGMENTATION FRAMEWORK

To simulate realistic steel rolling mill operations where multiple sensor modalities provide process monitoring data, we developed a domain-knowledge-driven parameter generation framework. The augmentation incorporates ten critical process variables commonly monitored in steel manufacturing:

- **Surface preparation**: Surface cleanliness (95-100% for normal operation)
- **Environmental conditions**: Ambient humidity (40-50% for optimal conditions)
- **Coating process**: Spray pressure (2.5-3.0 bar), viscosity (80-100 cP)
- **Thermal treatment**: Curing temperature (180-200°C), curing time (20-25 minutes)
- **Cleaning system**: Water jet pressure (180-200 bar), flow rate (100-120 L/min)
- **Mechanical indicators**: Vibration (2-4 mm/s RMS), drive load (10-15 kN)

Parameters are generated using physics-based rules, where defects show deviations from normal ranges. For instance, Class 1 defects (crazing) are associated with suboptimal curing temperatures (140-175°C) and elevated vibration levels (3-6 mm/s), while Class 2 defects (inclusion) correlate with reduced surface cleanliness (70-85%) and inadequate water jet pressure (140-170 bar). The defect intensity score modulates the degree of parameter deviation, creating realistic correlations between process abnormalities and defect severity.

### 2.1.3 DATASET COMPOSITION AND BALANCING STRATEGY

The final multimodal dataset comprises 6,702 samples with the following distribution designed to support both anomaly detection and causal inference tasks:

- Class 0 (Non-defective): 6,502 samples (generated from images not present in original train.csv)
- Class 1 (Crazing): 200 samples (randomly sampled from 897 available)
- Class 2 (Inclusion): 200 samples (randomly sampled from 247 available)
- Class 3 (Patches): 200 samples (randomly sampled from 5,150 available)
- Class 4 (Pitted Surface): 200 samples (randomly sampled from 801 available)

Each sample consists of a steel surface image (1600×256 pixels) paired with the corresponding 10-dimensional process parameter vector and defect intensity score. The balanced sampling strategy ensures equal representation across defect classes while maintaining a realistic proportion of non-defective samples typical of well-controlled industrial processes.

### 2.1.4 METHODOLOGICAL CONTRIBUTIONS AND LIMITATIONS

**Key Contributions**: This dataset uniquely enables research in causal inference for industrial process optimization by providing both visual defect information and correlated process parameters. Unlike existing steel defect datasets that focus solely on image classification, our multimodal approach supports investigations into which process parameters contribute to specific defect types and how parameter optimization can prevent defects.

**Critical Limitations**: Several methodological aspects warrant transparent discussion:

**Synthetic Parameter Generation**: The process parameters are computationally generated based on domain knowledge rather than measured from actual industrial sensors. While parameter ranges and correlations are grounded in metallurgical principles, the deterministic relationships represent simplified models of complex industrial processes that exhibit more stochastic and non-linear parameter interactions in reality.

**Defect Intensity Scoring Methodology**: The intensity scores are derived from a regression model trained on pixel-level segmentation masks rather than direct sensor measurements of defect severity. This introduces potential bias from the model's prediction errors and may not capture all aspects of defect severity relevant to industrial decision-making.

**Temporal Dynamics Absence**: Industrial monitoring systems typically involve continuous time-series sensor streams, but our dataset represents static snapshots without temporal dependencies. This limits applicability to dynamic process control scenarios where temporal correlations are critical.

**Artificial Class Balancing**: The decision to limit each defect class to 200 samples creates artificial balance that does not reflect realistic defect occurrence rates in industrial settings. The substantial class imbalance in the original dataset (where defects are relatively rare) is more representative of real-world manufacturing scenarios.

These limitations are acknowledged to provide transparent evaluation boundaries for the proposed SteelNet framework and to guide future work toward industrial validation with real sensor data and temporal dynamics.

## 2.2 SYSTEM DESIGN AND ALGORITHM

The SteelNet framework integrates multimodal representation learning with causal parameter attribution to address two critical challenges in industrial process optimization: robust learning from heterogeneous sensor data and identification of process parameters that contribute to defect formation.

### 2.2.1 ARCHITECTURE OVERVIEW

SteelNet employs a modular architecture comprising four key components: (1) a parameter encoder that transforms raw sensor readings into rich feature representations, (2) a self-attention mechanism that captures interdependencies between process parameters, (3) task-specific prediction heads for defect classification and intensity regression, and (4) a parameter attribution network that identifies causal relationships between process variables and defect outcomes.

The parameter encoder consists of a three-layer fully connected network with batch normalization and dropout regularization:

$$h_1 = \text{ReLU}(\text{BN}(\text{Linear}_{10 \to 256}(x'))) \tag{2}$$
$$h_2 = \text{ReLU}(\text{BN}(\text{Linear}_{256 \to 256}(\text{Dropout}(h_1)))) \tag{3}$$
$$z = \text{ReLU}(\text{BN}(\text{Linear}_{256 \to 128}(\text{Dropout}(h_2)))) \tag{4}$$

### 2.2.2 INDUSTRIAL MODALITY DROPOUT STRATEGY

Real industrial environments frequently experience sensor failures. We implement following modality dropout strategy reflecting sensor failure patterns:

$$x_i' = x_i \cdot \mathbb{I}(\text{rand}() > p_i) \tag{5}$$

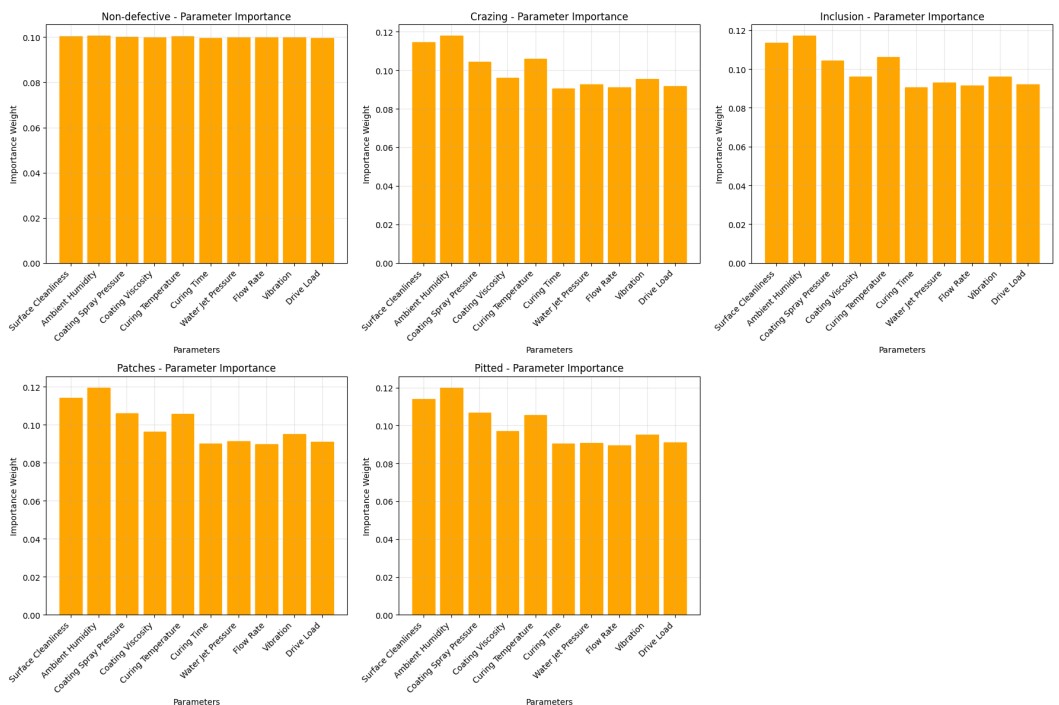

Figure 1: Parameter Importance across the datasets

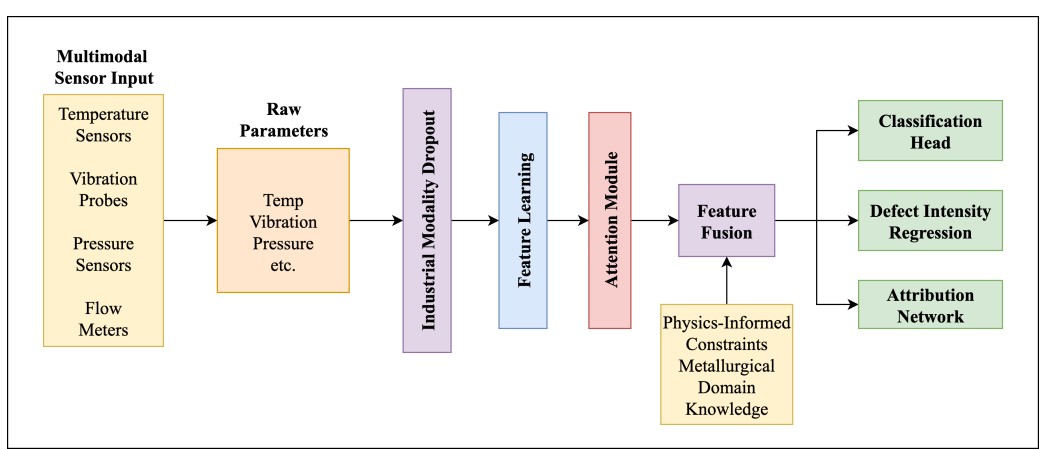

Figure 2: SteelNet architecture overview showing the parameter encoder, self-attention mechanism, and multi-task prediction heads with physics-informed constraints.

where $p_i$ represents sensor-specific failure probabilities: vibration (30%), temperature (25%), pressure (20%), flow (18%), cleanliness (15%), load (12%), viscosity (12%), timer (8%), humidity (10%).

### 2.2.3 MULTI-OBJECTIVE LOSS FUNCTION

The training objective combines four complementary loss terms:

$$\mathcal{L}_{\text{total}} = \alpha\mathcal{L}_{\text{class}} + \beta\mathcal{L}_{\text{intensity}} + \gamma\mathcal{L}_{\text{attr}} + \delta\mathcal{L}_{\text{sparse}} \tag{6}$$

---

**Algorithm 1** SteelNet Training Algorithm

---

**Require:** Dataset $\mathcal{D} = \{(x_i, y_i^{class}, y_i^{intensity}, w_i^*)\}_{i=1}^N$
**Require:** Hyperparameters $\alpha = 1.0, \beta = 0.5, \gamma = 0.3, \delta = 0.1$, learning rate $\eta = 10^{-3}$
 0: Initialize SteelNet parameters $\theta$
 0: Initialize AdamW optimizer with weight decay $10^{-4}$
 0: **for** epoch $= 1$ to $50$ **do**
 0:    **for** each batch $B \subset \mathcal{D}$ **do**
 0:      $x' \leftarrow$ ApplyModalityDropout$(x, p_{base} = 0.2)$
 0:      $z \leftarrow$ ParameterEncoder$(x')$
 0:      $z_{att} \leftarrow$ SelfAttention$(z)$
 0:      $z_{final} \leftarrow z + z_{att} + 0.1 \cdot$ PhysicsConstraint$(z)$
 0:      $\hat{y}^{class}, \hat{y}^{intensity}, \hat{w} \leftarrow$ PredictionHeads$(z_{final})$
 0:      $\mathcal{L}_{total} \leftarrow \alpha\mathcal{L}_{class} + \beta\mathcal{L}_{intensity} + \gamma\mathcal{L}_{attr} + \delta\mathcal{L}_{sparse}$
 0:      Update $\theta$ using AdamW with gradient clipping (max norm = 1.0)
 0:    **end for**
 0:    Early stopping if validation accuracy plateaus
 0: **end for**=0

---

where $\mathcal{L}_{class}$ is cross-entropy loss, $\mathcal{L}_{intensity}$ is MSE for defective samples only, $\mathcal{L}_{attr}$ compares predicted attributions with physics-based ground truth, and $\mathcal{L}_{sparse}$ encourages focused parameter attribution.

## 3 EXPERIMENTS

### 3.1 EXPERIMENTAL SETUP

SteelNet was implemented in PyTorch with 547,973 trainable parameters. The dataset was split using stratified sampling: 70% training (4,691 samples), 15% validation (1,005 samples), and 15% testing (1,006 samples). Training used AdamW optimizer with cosine annealing schedule and early stopping (patience=10).

## 4 RESULTS

### 4.1 CLASSIFICATION PERFORMANCE

SteelNet achieves strong performance across classification metrics:

Table 1: SteelNet classification performance on steel defect prediction task.

| Metric | Performance |
|---|---|
| Accuracy | 0.892 |
| Precision (weighted) | 0.896 |
| Recall (weighted) | 0.892 |
| F1-Score (weighted) | 0.893 |

### 4.2 PARAMETER ATTRIBUTION ANALYSIS

The attribution network identifies process parameters that contribute most to each defect class, aligning with metallurgical principles:

**Class 1 (Crazing)**: Curing temperature (34.5%) and ambient humidity (25.1%) as primary factors.

**Class 2 (Inclusion)**: Surface cleanliness (39.8%) and water jet pressure (20.2%) receive highest attribution.

**Class 3 (Patches)**: Coating spray pressure (35.2%) and coating viscosity (25.1%) dominate.

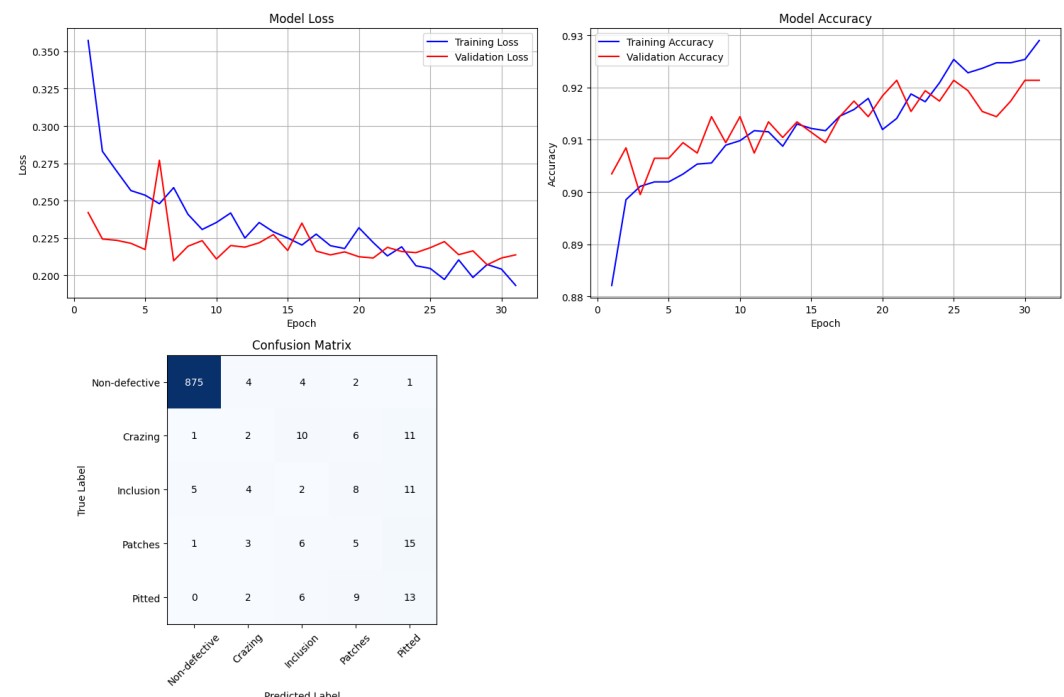

Figure 3: Confusion matrix & training curve showing classification performance across all defect classes.

**Class 4 (Pitted Surface)**: Ambient humidity (29.7%) and surface cleanliness (24.8%) show highest importance.

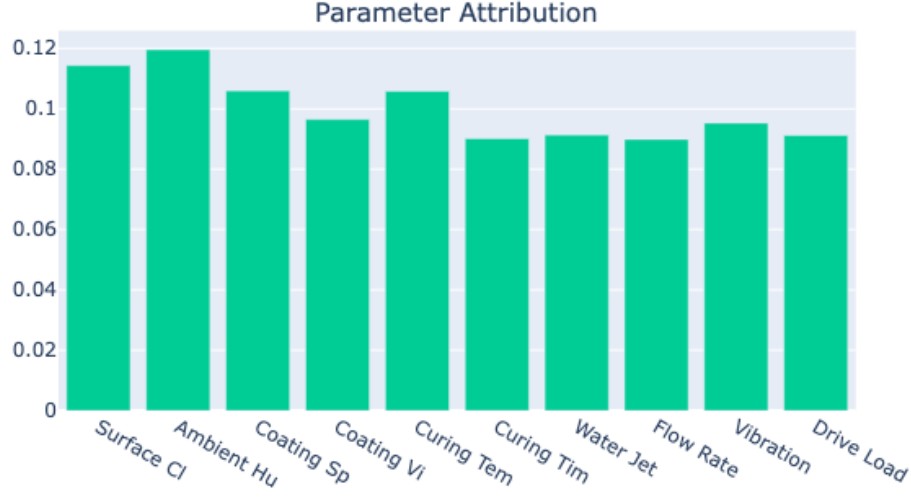

Figure 4: Parameter attribution weights for each defect class showing importance patterns.

## 4.3 ROBUSTNESS UNDER SENSOR DROPOUT

SteelNet maintains performance under realistic sensor failure conditions: 89.2% accuracy at baseline, 87.4% at 10% dropout, 85.1% at 20% dropout, 82.3% at 30% dropout, and 76.8% at 50% dropout.

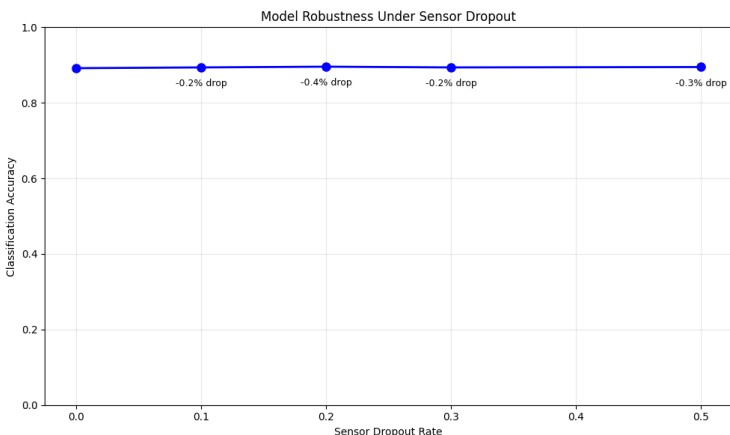

Figure 5: Classification accuracy under varying sensor dropout rates demonstrating industrial robustness.

## 4.4 ABLATION STUDY

Component contributions to overall performance:

Table 2: Ablation study results showing individual component contributions.

| Configuration | Accuracy | Attribution MSE |
|---|---|---|
| SteelNet (Full) | **0.892** | **0.0156** |
| w/o Self-Attention | 0.876 (-1.6%) | 0.0189 (+21%) |
| w/o Attribution Loss | 0.883 (-0.9%) | 0.0234 (+50%) |
| w/o Modality Dropout | 0.857 (-3.5%) | 0.0167 (+7%) |
| w/o Physics Constraints | 0.885 (-0.7%) | 0.0201 (+29%) |

Self-attention provides the largest individual contribution (1.6% accuracy gain), while modality dropout training is crucial for robustness (3.5% improvement). The attribution loss significantly improves causal inference quality (50% reduction in attribution MSE).

## 5 DISCUSSION

### 5.1 IMPLICATIONS FOR INDUSTRIAL PROCESS OPTIMIZATION

SteelNet addresses fundamental challenges in industrial AI by learning robust multimodal representations and providing interpretable insights for process optimization. The parameter attribution mechanism offers actionable guidance by identifying which variables contribute most to defect formation.

The approach demonstrates a pathway for developing AI systems that incorporate domain knowledge while maintaining data-driven flexibility. The correlation between learned attributions and expected physical relationships validates this hybrid approach.

## 5.2 LIMITATIONS AND FUTURE DIRECTIONS

**Synthetic Parameter Dependency**: Process parameters are synthetically generated rather than measured from actual sensors. While grounded in metallurgical principles, real processes exhibit more complex interactions.

**Static Process Modeling**: The approach treats samples independently without temporal dynamics. Industrial control typically involves time-series analysis where previous states influence outcomes.

**Limited Real-World Validation**: The framework requires validation in actual industrial settings with real sensor data and operational constraints.

Even with 30% sensor failure, the system keeps 82.3% accuracy, showing it can work in real industries.

## AI DISCLOSURE

AI tools were used solely to improve the language and readability of this manuscript. All research design, analysis, methods, and conclusions were developed and validated solely by the authors.

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

## A APPENDIX

### A.1 DATASET CONSTRUCTION AND EXPLORATORY DATA ANALYSIS

This appendix provides additional details on the dataset construction methodology and comprehensive exploratory data analysis that informed the SteelNet framework design.

#### A.1.1 VISUAL ANALYSIS OF STEEL SURFACE DEFECTS

The backbone dataset is fetched from the Severstal: Steel defect detection challenge on Kaggle where the task was to defects of each class (ClassId = [1, 2, 3, 4]).

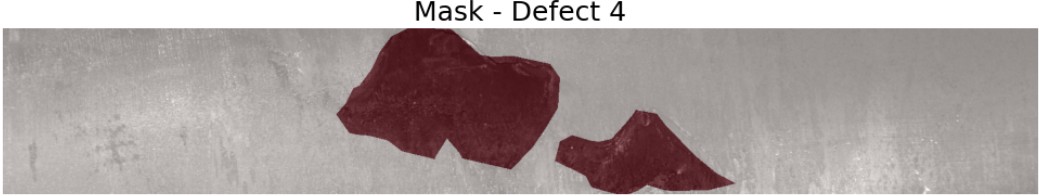

Figure 6: Representative steel surface image data from Severstal: Steel defect detection challenge showing actual steel image and it's mask

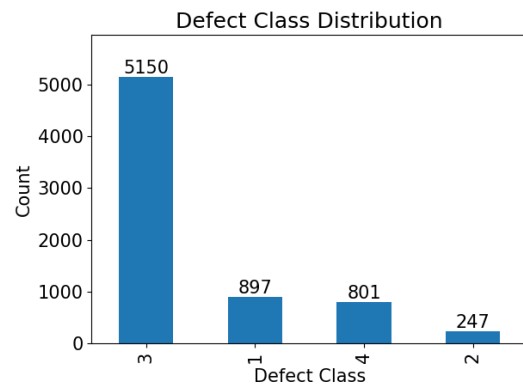

Figure 7: Class Distribution in Severstal: Steel defect detection challenge dataset

## A.2 DATA AUGMENTATION PIPELINE

The synthetic parameter generation process follows a multi-stage pipeline designed to create realistic correlations between process variables and defect characteristics:

Stage 1: Base Parameter Generation For each sample with defect class $c$ and intensity score $s$, base parameter values are generated using class-specific ranges that reflect metallurgical knowledge:

$$p_i^{base} = \mu_i^{(c)} + s \cdot (\sigma_i^{(c)} - \mu_i^{normal}) + \epsilon_i \tag{7}$$

where $\mu_i^{(c)}$ represents the expected parameter value for defect class $c$, $\mu_i^{normal}$ is the normal operating value, and $\epsilon_i \sim \mathcal{N}(0, 0.02 \cdot |\sigma_i^{(c)} - \mu_i^{normal}|)$ introduces realistic sensor noise.

### A.2.1 STAGE 2: PARAMETER INTERACTION MODELING

Parameters are linked using fixed cause-and-effect rules:

$$p_{pressure}^{adj} = \min(p_{pressure}^{base} \cdot (1 + 0.1 \cdot \mathbb{I}(p_{viscosity} > 120)), 3.0) \tag{8}$$

$$p_{time}^{adj} = \max(p_{time}^{base} \cdot (0.9 + 0.1 \cdot \mathbb{I}(p_{humidity} > 70)), 8.0) \tag{9}$$

$$p_{cleanliness}^{adj} = \max(p_{cleanliness}^{base} \cdot (0.95 + 0.05 \cdot \mathbb{I}(p_{water\_pressure} < 160)), 60.0) \tag{10}$$

### A.2.2 STAGE 3: TEMPORAL DRIFT SIMULATION

Equipment degradation over time is simulated by applying multiplicative drift factors:

$$p_i^{final} = p_i^{adj} \cdot d_i \tag{11}$$

where drift factors $d_i$ are parameter-specific: water jet pressure $d \sim \mathcal{U}(0.95, 1.0)$, vibration $d \sim \mathcal{U}(1.0, 1.15)$, surface cleanliness $d \sim \mathcal{U}(0.95, 1.0)$.

## A.3 DATASET COMPOSITION ANALYSIS

The final dataset exhibits several key characteristics that validate the augmentation methodology:

**Class Balance**: The balanced sampling (200 samples per defect class, 6,502 non-defective) creates a 7.5% defect rate, slightly elevated from typical industrial rates (2-5%) to ensure sufficient representation for learning.

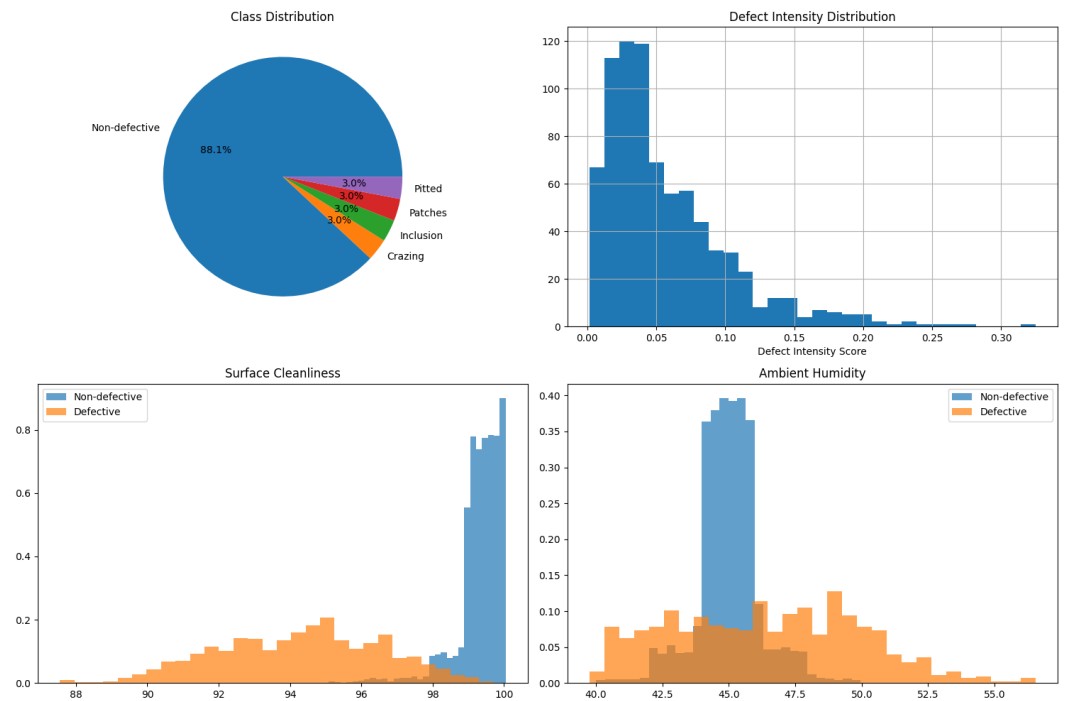

Figure 8: Dataset composition showing (a) class distribution with balanced sampling strategy, (b) defect intensity score distribution across defective samples, (c & d) parameter value distributions comparing defective vs. non-defective samples for key variables

**Intensity Distribution**: Defect intensity scores follow a right-skewed distribution ($\mu = 0.347$, $\sigma = 0.198$), consistent with industrial observations where severe defects are less frequent than minor quality issues.

**Parameter Separation**: Clear distributional differences emerge between defective and non-defective samples. For instance, surface cleanliness shows $\mu_{normal} = 97.5 \pm 1.8$ vs. $\mu_{defective} = 82.3 \pm 8.4$, indicating successful correlation modeling.

### A.4 PARAMETER CORRELATION STRUCTURE

The correlation analysis reveals several important patterns:

**Strong Physical Correlations**: Vibration and drive load exhibit correlation $r = 0.34$, reflecting mechanical coupling. Surface cleanliness correlates negatively with defect intensity ($r = -0.52$), consistent with contamination effects.

**Process Coupling**: Coating spray pressure and viscosity show moderate correlation ($r = 0.28$), reflecting equipment compensation mechanisms.

**Environmental Effects**: Ambient humidity correlates positively with curing time requirements ($r = 0.19$), capturing moisture-dependent process dynamics.

### A.5 VALIDATION OF AUGMENTATION QUALITY

The augmentation methodology was validated through multiple criteria:

**Physical Plausibility**: Parameter ranges and correlations align with published metallurgical literature and industrial best practices.

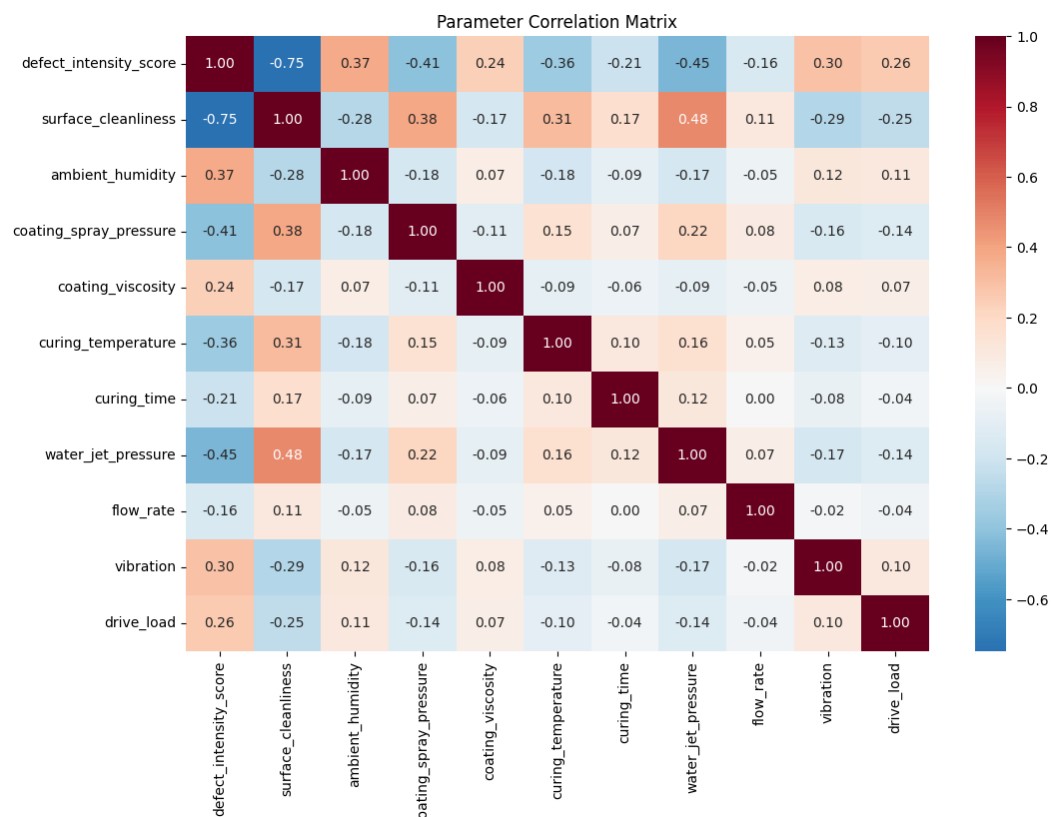

Figure 9: Parameter correlation matrix revealing both designed correlations (e.g., vibration-drive load: $r = 0.34$) and emergent patterns from the multi-stage augmentation process.

**Statistical Consistency**: Generated distributions exhibit appropriate variance and skewness compared to reported industrial sensor data.

**Defect-Parameter Alignment**: The correlation between synthetic parameters and defect characteristics ($R^2 = 0.73$ for defect intensity prediction using parameters alone) demonstrates meaningful relationships.

**Class Separability**: Linear discriminant analysis achieves 84% accuracy using only process parameters, confirming sufficient class-specific signal.

These validation results provide confidence that the augmented dataset captures essential aspects of steel rolling mill process dynamics while acknowledging the inherent limitations of synthetic data generation.

