# OpenReview forum: "SteelNet: Multimodal Representation Learning for Industrial Process Optimization"
_ICLR.cc/2026/Conference — Submitted to ICLR 2026_

### Official Review · Reviewer_Mdjv · 2025-10-23

**Soundness:** 2
**Presentation:** 2
**Contribution:** 1
**Rating:** 2
**Confidence:** 4

**Summary:**

This study proposes SteelNet, a multimodal learning framework that jointly uses images and sensor data to predict defect types and analyze their underlying causes on steel surfaces. Specifically, SteelNet integrates an image encoder and a sensor encoder, where a classification head predicts defect types and an attribution network effectively identifies the contributions of multiple sensors. Moreover, the framework uses modality dropout and incorporates physics-informed constraints to address the modality-absence problem in steel manufacturing processes. Experimental results demonstrate that SteelNet achieves robust performance and provides interpretable insights into the causal relationships between sensors and defects.

**Strengths:**

1) The proposed framework extends multimodal modeling to the steel domain for defect detection and its interpretability, which are crucial in steel manufacturing.
2) This study defines and addresses the modality-absence problem, which has not been considered in previous research. To tackle this issue, the study uses dropout and effectively simulates missing modalities.
3) To demonstrate the effectiveness of SteelNet, this study generated simulation data that were designed to reflect real-world conditions, including sensor-absence scenarios where certain sensors were not recorded or became unavailable.

**Weaknesses:**

[Method]
1) This method does not meet the standards for publication at a top-tier AI conference. While the attempt to integrate multimodal representation learning with causal interpretability is conceptually sound, the proposed components such as the parameter encoder, multi-task prediction heads, and the attribution loss with sparsity regularization rely on widely used techniques from existing studies. Moreover, the physics-informed constraint is formulated as a simple supervised learning framework rather than offering a novel algorithmic innovation.
2) The description of the method section is insufficient and requires more detailed explanation. For example, definitions of the attribute loss and the sparsity loss are missing.

[Experiments & Results]
1) The proposed method is not compared with existing studies. Although it achieves relatively high accuracy, it is difficult to determine whether the model truly outperforms prior approaches without comparative evaluation.
2) The paper does not include experiments on real-world data. Although these limitations are clearly stated, it is difficult to validate real-world applicability using solely generated dataset. Moreover, the comparison between real and simulated data mentioned in Appendix relies only on a few statistical indicators, making it somewhat weak. The authors should either (1) add experiments based on real data, or (2) provide stronger evidence that the simulated data is indistinguishable from real data.
3) The explanations for each figure and table in results are insufficient. For example, there is no detailed analysis or discussion related to Figure 3, and Figure 4 does not specify which defect class it represents. Providing clearer and more detailed explanations for each figure and table would greatly improve the readers’ understanding of the paper.

**Questions:**

1) Why did the proposed method use the predicted values from a separately trained ResNet model as intensity labels instead of using the actual intensity values? Using model-predicted values as labels seems inefficient and may introduce additional sources of error. A more detailed explanation of this design choice is needed.
2) Why was the proposed method validated using an equally sampled dataset rather than the original data distribution? In real-world scenarios, the proportions of different defect types are typically imbalanced. Such uniform sampling may hinder a realistic comparison with actual industrial conditions. Therefore, it is recommended to also include experimental results based on the original class distribution.

---

### Official Review · Reviewer_KXx5 · 2025-10-29

**Soundness:** 2
**Presentation:** 2
**Contribution:** 2
**Rating:** 2
**Confidence:** 5

**Summary:**

The paper proposes SteelNet, a multimodal representation learning framework aimed at improving industrial process optimization， specifically in steel rolling mills. It integrates cross-modal alignment and a modality-dropout strategy to learn robust representations from heterogeneous sensor data (e.g., vibration, temperature, pressure). The model also incorporates a parameter attribution mechanism for causal interpretation of process variables and their relation to defects. The authors construct a synthetic multimodal dataset by extending the Severstal Steel Defect Detection dataset with simulated process parameters to evaluate the framework’s performance under missing modalities.

**Strengths:**

1. SteelNet addresses real-world challenges in industrial environments such as missing sensor data, noise, and cross-modal inconsistencies, which enhances its industrial applicability.

2. The framework integrates multiple components (cross-modal alignment, self-attention, causal attribution) and demonstrates solid performance with clear ablation and robustness studies.

**Weaknesses:**

1. The paper mainly applies existing multimodal learning ideas (cross-modal fusion, modality dropout, attention, attribution) to an industrial context. It does not introduce a fundamentally new algorithmic contribution, which fails to meet ICLR’s originality and theoretical novelty standards.

2. Evaluation is conducted only on a synthetically augmented version of a single dataset, without evidence on broader industrial or real sensor datasets, limiting the generalizability.

3. Some plots (e.g., architecture overview and confusion matrices) are too rough or visually cluttered, lacking publication-level polish expected for ICLR.

**Questions:**

See Weaknesses

---

### Official Review · Reviewer_ow7r · 2025-10-31

**Soundness:** 2
**Presentation:** 2
**Contribution:** 1
**Rating:** 2
**Confidence:** 2

**Summary:**

The paper introduces SteelNet, a multimodal representation learning approach designed for defect detection in steel rolling mills.
The method integrates multiple data sources and aims to identify the variables that contribute most to defect occurrence.
To achieve this, it employs a synthetic parameter generation process intended to reproduce known domain-specific cause–effect relationships and environmental correlations.

**Strengths:**

- The paper is overall well-written and easy to follow.
- The synthetic parameter generation mechanism is conceptually interesting, as it attempts to encode domain knowledge and known physical correlations into the learning process.
- The integration of multimodal data and the focus on explainability (via variable contribution analysis) are valuable contributions to industrial AI and fault diagnosis applications.

**Weaknesses:**

- The experimental validation is limited to a single dataset, making it difficult to assess the generality and broader applicability of the proposed method.
- Although the authors claim robustness to missing data and early detection of critical events, no quantitative study or sensitivity analysis is presented to substantiate these claims.
- Some methodological choices, such as disregarding temporal dependencies and class imbalance, raise concerns about the validity of the reported results. Other aspects also seem ad hoc or insufficiently detailed (e.g., empirically determined weights).

**Questions:**

1) Are the empirically determined weights fixed, or are they updated during training or fine-tuning?
2) Table 1 reports performance metrics, but it is unclear whether these values refer to a specific defect class or represent averages across all classes. Could the authors clarify this?
3) How robust is the model to missing data or corrupted modalities?
4) What is the timeliness or early-warning performance of the model in detecting faults?
5) How representative is the dataset used? If class distributions differ from real-world conditions, how might this affect the model’s deployment validity?
6) Given that this type of process is inherently sequential, how effective is the proposed approach without modeling temporal dependencies? Is there any evaluation of the model’s ability to detect faults early, i.e., before failure events occur?
7) The authors claim that the approach applies to other industrial settings. What aspects are specifically tailored to steel rolling mills, and what components are generalizable? Dataset characteristics, domain-specific feature engineering, or architectural design?

---

### Official Review · Reviewer_hZkH · 2025-11-01

**Soundness:** 1
**Presentation:** 1
**Contribution:** 1
**Rating:** 2
**Confidence:** 4

**Summary:**

This paper introduces SteelNet, a multimodal representation-learning framework for steel-mill monitoring that aims to stay robust when sensors fail and to surface actionable parameter-level attributions for defects. It pairs a new (partly synthetic) multimodal dataset with a model that uses a parameter encoder, self-attention, physics-informed constraints, modality dropout, and multi-task heads for defect class and “intensity” prediction, plus an attribution head to highlight root-cause variables.

**Strengths:**

1. Integration of industrial requirements with AI design:
The paper effectively connects real industrial problems with AI methods. Instead of building a generic model, it designs the system to handle real issues in steel manufacturing, such as missing sensor data, few labeled samples, and the need for clear explanations of defect causes. This focus on practical challenges makes the work more useful and relevant for industry applications, not just academic research.

2. Systematic ablation study:
The authors conduct an ablation analysis that clearly disentangles the effect of each model component — self-attention, modality-dropout, attribution loss, and physics constraints.

3. Acknowledgment and discussion of limitations:
The paper clearly states its main limitations — including the use of synthetic data, simplified treatment of time, and absence of real factory deployment. This honest discussion shows that the authors understand the boundaries of their work and helps readers see what still needs to be tested in real conditions.

**Weaknesses:**

1. Figures: low readability and polish (plus a minor mistake):
Several plots are hard to read (small fonts, low contrast, dense legends) and look inconsistent across sections (e.g., architecture/importance figs; robustness and appendix plots). Clearer typography, consistent palettes, and vector graphics would help. Also fix the caption typo “it’s mask” → “its mask” in Fig. 6.

2. System/algorithm description feels sketch-level rather than fully academic:
While Fig. 2 and “Algorithm 1” give a high-level view, key pieces remain under-specified—especially the exact fusion with the image modality and the formal definition of the physics constraint (it appears as a black-box term PhysicsConstraint(z)). Please spell out the fusion pathway, units/normalization, and the constraint’s closed form (including any coefficients) so others can replicate precisely.

3. Heavy reliance on synthetic parameters and static samples limits external validity:
The paper itself notes that process parameters are synthetically generated, time dynamics are omitted, and there is no validation on real sensor data. This makes generalization to real plants uncertain. A small real-sensor/time-series study (even a pilot) would strengthen the claims.

4. Claims not fully supported as contributions:
The paper states it targets equipment availability and improves operational decision-making, but the experiments do not measure uptime/availability or control benefits; as written, that should not be counted as a demonstrated contribution. Also, the defect-intensity scoring (ResNet-34 trained on masks) is a useful dataset step, but not methodically novel—please frame it as dataset preparation rather than a core algorithmic contribution.

**Questions:**

Q1. Figures / readability:
Could you regenerate all figures with vector graphics, larger fonts, clear axis units, consistent color palettes, and readable legends—especially Figs. 5–9? (Also, please fix the caption typo “it’s mask” → “its mask” in Fig. 6.)

Q2. Method clarity: fusion + physics constraint:
How exactly are image features fused with process-parameter embeddings, and what is the closed-form definition (with coefficients) of the physics-constraint term used in training? The ablation refers to “w/o Physics Constraints” but the constraint itself isn’t fully specified.

Q3. Data realism and leakage control:
Given the reliance on synthetic parameters and static samples, can you provide any validation on real sensor time-series—or, at minimum, describe safeguards against leakage between the intensity model and the parameter generation pipeline?

Q4. Baselines and statistical rigor:
Beyond ablations, will you add head-to-head baselines (e.g., strong tabular models and established missing-modality methods), report mean±std over multiple seeds, and clarify the sensor-dropout protocol (failure sampling pattern, seeds, confidence intervals)?

---

### Meta-Review · Area_Chair_hx4p · 2026-01-06

**Summary:**

The submission targets a practically important industrial monitoring setting (multimodal sensors + images; missing-modality robustness; interpretability).

However, reviewers converged on (i) limited novelty beyond assembling standard multimodal components, (ii) under-specified method details (notably image parameter fusion and the physics constraint term), and (iii) insufficient empirical support. Evaluation is largely on a single, synthetically augmented dataset with simplified/non-temporal assumptions. Overall, the evidence and clarity are not strong enough to support acceptance.

**Reviewer Concerns:**

There was no rebuttal provided.

Outstanding concerns:
* Precise specification of the fusion pathway
* Clear definition of closed-form physics constraint
* stronger baselines (eg. missing modality, and tabular models)
* validation beyond one synthetic dataset (ideally real or time series data)
* clearer handling of class imbalance

**Reviewer Scores:**

n/a

---

### Decision · Program_Chairs · 2026-01-26

Reject